# Genetic diversity, genetic population structure and epidemiology of multidrug resistance *Neisseria gonorrhoeae* from Kenya

**Henry Kiema Musee[1], Dennis Wamalabe Mukhebi[1,2]\***

1 Department of Biochemistry and Biotechnology, Pwani University, Kilifi, Kenya, 2 Pwani University Biosciences Research Centre (PUBReC), Pwani University, Kilifi, Kenya

\* mukhebidennis95@gmail.com, d.mukhevi@pu.ac.ke

## Abstract

*Neisseria gonorrhoeae* is the bacterial agent responsible for gonorrhea, a common sexually transmitted infection. The emergence of *Neisseria gonorrhoeae* multidrug-resistant (MDR) strains a presents a critical public health threat, especially due to its contribution to antimicrobial resistance (AMR) and treatment failure. Currently, resistance profiling of *N. gonorrhoeae* relies on phenotypic methods such as minimum inhibitory concentration (MIC) testing and identification of known resistance mutations. These are with limited application of genome-wide approaches to understand resistance evolution. The lack of genomic epidemiology data among Low- and Middle-Income Countries (LMICs) regions such as Kenya, hampers effective AMR tracking and designation of evidence-based, targeted treatments. This study aims to investigate the genetic diversity, population structure, and recombination dynamics of MDR *N. gonorrhoeae* isolates from Kenya using whole-genome SNP analysis. A total of 92 genomes (72 FASTQ reads and 20 assembled genomes) were retrieved from NCBI. De novo assembly, identification of AMR genes and variant calling were conducted, followed by Principal Component Analysis (PCA), Neighbor-Net clustering, nucleotide diversity ($\pi$), and linkage disequilibrium (LD) decay analysis to assess population structure and recombination patterns. Our results revealed a predominant genetic cluster with several divergent outlier strains, indicating moderate population differentiation. Despite the lack of strong geographic separation in the overall genomic structure, significant regional differences were observed in antimicrobial resistance gene burden. Western Kenyan regions (Kisumu and Kombewa) exhibited higher AMR gene counts despite genetic similarity to isolates from other regions, suggesting that local antibiotic selection pressures, rather than population isolation, are driving the accumulation of resistance determinants. Across the genome, nucleotide diversity was variable with distinct recombination hotspots. A rapid LD decay within the first 1000 bp suggested a high overall recombination rate. These results indicate that recombination plays a pivotal role in shaping genetic variability and AMR

**Data availability statement:** The raw whole genome sequences of Neisseria gonorrhoeae used in this project were deposited in the NCBI GenBank under BioProject PRJNA481622 (for the data which had only reads 72) and PRJNA660404 (for 10 genomes from Nairobi, Kenya).

**Funding:** The author(s) received no specific funding for this work.

**Competing interests:** The authors declare no conflict of interest.

evolution in *N. gonorrhoeae* populations. The absence of strong geographic structure further implies that transmission dynamics, rather than regional isolation, drive the spread of resistance. In conclusion, genome-wide SNP analysis offers valuable insights into the genetic diversity and populations structure of MDR *N. gonorrhoeae* in Kenya. These findings support the integration of genomic surveillance into national strategies for antimicrobial resistance control. This study reveals how genetic analysis can guide better strategies to track and control drug-resistant gonorrhea in Kenya.

## Introduction

Gonorrhea, caused by the bacterium *Neisseria gonorrhoeae*, is a common sexually transmitted infection that, if untreated, can lead to severe complications such as pelvic inflammatory disease (PID) and disseminated gonococcal infection (DGI) [1]. Furthermore, the public health threat posed by *N. gonorrhoeae* is compounded by high rates of co-infection with other sexually transmitted infections (STIs), including HIV and mpox virus. This can exacerbate disease severity, complicate diagnosis and negatively affect treatment outcomes [2]. Over the past 70 years, *Neisseria gonorrhoeae* has progressively developed resistance to several antibiotic classes, including penicillin, aminoglycosides, macrolides, and tetracycline, which were once effective treatment options. The declining efficacy of these antibiotics is largely attributed to genetic mutations within *Neisseria gonorrhoeae* genome. Notably, resistance to tetracycline has been reported, likely exacerbated by the discontinuation of these drugs as a primary treatment option ([3]. The introduction of ciprofloxacin and other fluoroquinolones was initially effective in controlling *Neisseria gonorrhoeae* infection [4]. However, due to the emergence of bacterial resistance, fluoroquinolones, including ciprofloxacin, are no longer considered reliable treatment options [5].

Resistance to ciprofloxacin and other fluoroquinolones has necessitated the use of third-generation cephalosporins, such as cefixime and ceftriaxone, due to their high efficacy even at lower doses. These antibiotics were initially recommended as the first-line treatment for *Neisseria gonorrhoeae* infections. However, despite their effectiveness, the bacterium has developed resistance mechanisms that compromise their efficacy. The global dissemination of resistant *Neisseria gonorrhoeae* strains has led to a significant public health challenge, with resistance spreading rapidly across populations. In Kenya, resistance among most available antibiotics to *Neisseria gonorrhoeae* isolates has been linked to mutations in the gyrA and parC genes, which confer fluoroquinolone resistance [6]. A study by Juma et al., [7] uncovered that ciprofloxacin resistance is associated with multiple mutations in both gyrA and parC. Despite studies documenting resistance through phenotypic and genotypic methods, a significant knowledge gap remains regarding the genomic mechanisms of AMR in African LMICs like KenyaWhile conventional diagnostic approaches rely on culture and microscopy, the field is rapidly adopting molecular testing, including nucleic acid amplification tests (NAATs) and emerging

technologies like isothermal nucleic acid amplification (e.g., TMA), to enable rapid and decentralized detection of *N. gonorrhoeae* [8].

In this study we conducted a genomic analysis of 56 *Neisseria gonorrhoeae* isolates from Kenya, all of which demonstrated antimicrobial resistance to more than two antibiotics. The research aimed at elucidating genetic diversity, evolutionary features, and the potential impact of epidemiological factors using genome-wide SNP analysis. These factors include geographical distribution, transmission dynamics, antibiotic usage patterns, and host demographics, on the development of antimicrobial resistance.

The 92 high quality genomes (72 FASTQ reads and 20 assembled genomes) were retrieved from NCBI. De novo assembly was applied to the reads and then subjected to the SNP calling. Principal Component Analysis (PCA), Neighbor-Net analysis, nucleotide diversity ($\pi$) and Linkage disequilibrium (LD) decay plots were used to characterize the genetic relationships, diversity and recombination. These resulted into detecting clustering patterns, evolutionary divergence, and recombination hotspots relevant to resistance evolution.

Our findings revealed a dominant genetic cluster among Kenyan isolates, accompanied by several genetically divergent outliers, which indicates a moderate population differentiation. The LD decay and nucleotide diversity analyses provided strong evidence of frequent recombination events, suggesting that genetic exchange plays a pivotal role in shaping resistance traits. Moreover, the absence of strong geographic structuring highlights the importance of transmission dynamics over regional isolation in driving resistance spread.

This study advances the understanding of how the genetic plasticity and recombination influences resistance among *Neisseria gonorrhoeae* populations. Therefore, it emphasizes on the need for genomic surveillance in the development of targeted, evidence-based interventions. Our research also demonstrates the value of genome-wide SNP data in informing public health policy and highlights the urgency of integrating genomic tools into AMR monitoring strategies in LMICs such as Kenya. Genomic surveillance, as performed in this study, remains crucial to complement these rapid diagnostic tools by tracking the emergence and spread of antimicrobial resistance (AMR) mechanisms.

## Materials and methods

### Genomic data retrieval

A total of 72 FASTQ reads were obtained from the NCBI database under BioProject accession PRJNA481622, along with 20 complete genome sequences retrieved from BioProject PRJNA660404.

### Processing and determination of variants

The quality of the raw sequencing reads was evaluated using FastQC version 0.11.9. The low-quality bases with a Phred score below 20 and adapter sequences were trimmed using Trimmomatic PE v0.39 [9]. The cleaned reads underwent de novo assembly using SPAdes v4.0.0 [10], with default assembly parameters. The genome quality was evaluated using QUAST version 5.2 [11], and BUSCO tool [12], to determine genome completeness, (S1 File), with the neisseriales_odb10 database. The number of predicted genes and their functions were determined to identify genes associated with antibiotic resistance (Table 1). The assembled genomes were aligned to a reference genome (GenBank assembly number GCA_013030075.1) using Burrows-Wheeler Aligner (BWA) v0.7.18-r1243 [13]. The reference genome was first indexed using bwa index command and samtools faidx. Each genome was aligned to the reference, by creating a batch processing script which generated a Sequence Alignment Map (SAM) for each genome. The SAM files were converted to Binary Alignment Map (BAM) and sorted using SAMtools v1.15.1 [14]. All sorted genomes were merged and indexed using samtools. Bcftools v1.16 mpileup command was used to call for the variants using the reference genome and the merged sorted bam file, to produce Variants.vcf file, which contained the variants. Low quality variants (QUAL< 20) were filtered out using bcftools vcftools v0.1.16 [15].

**Table 1. Number of genes (amr) per sample. The distribution of AMR genes across all the retrieved genomes from NCBI.**

| Sample | Number of respective genes in each sample | | | | | | | | | | | |
|---|---|---|---|---|---|---|---|---|---|---|---|---|
| | TEM-1 | TEM-104 | TEM-206 | farA | farB | lnuA | macA | macB | mtrC | mtrD | mtrE | tetM |
| GCA_014844795.1_ASM1484479v1_genomic.fna | 0 | 0 | 0 | 1 | 1 | 0 | 1 | 1 | 1 | 1 | 1 | 0 |
| GCA_014844815.1_ASM1484481v1_genomic.fna | 0 | 0 | 0 | 1 | 1 | 0 | 1 | 1 | 1 | 1 | 0 | 0 |
| GCA_014844835.1_ASM1484483v1_genomic.fna | 0 | 0 | 0 | 1 | 1 | 0 | 1 | 1 | 1 | 1 | 1 | 0 |
| GCA_014844855.1_ASM1484485v1_genomic.fna | 0 | 0 | 0 | 1 | 1 | 0 | 1 | 1 | 1 | 1 | 1 | 0 |
| GCA_014844875.1_ASM1484487v1_genomic.fna | 0 | 0 | 0 | 1 | 1 | 0 | 1 | 1 | 1 | 1 | 1 | 0 |
| GCA_014844895.1_ASM1484489v1_genomic.fna | 0 | 0 | 0 | 1 | 1 | 0 | 1 | 1 | 1 | 1 | 1 | 0 |
| GCA_014844915.1_ASM1484491v1_genomic.fna | 0 | 0 | 0 | 1 | 1 | 0 | 1 | 1 | 1 | 1 | 1 | 0 |
| GCA_014844935.1_ASM1484493v1_genomic.fna | 0 | 0 | 0 | 1 | 1 | 0 | 1 | 1 | 1 | 1 | 1 | 0 |
| GCA_014844955.1_ASM1484495v1_genomic.fna | 0 | 0 | 0 | 1 | 1 | 0 | 1 | 1 | 1 | 1 | 1 | 0 |
| GCA_014844975.1_ASM1484497v1_genomic.fna | 0 | 0 | 0 | 1 | 1 | 0 | 1 | 1 | 1 | 1 | 1 | 0 |
| GCF_014844795.1_ASM1484479v1_genomic.fna | 0 | 0 | 0 | 1 | 1 | 0 | 1 | 1 | 1 | 1 | 1 | 0 |
| GCF_014844815.1_ASM1484481v1_genomic.fna | 0 | 0 | 0 | 1 | 1 | 0 | 1 | 1 | 1 | 1 | 0 | 0 |
| GCF_014844835.1_ASM1484483v1_genomic.fna | 0 | 0 | 0 | 1 | 1 | 0 | 1 | 1 | 1 | 1 | 1 | 0 |
| GCF_014844855.1_ASM1484485v1_genomic.fna | 0 | 0 | 0 | 1 | 1 | 0 | 1 | 1 | 1 | 1 | 1 | 0 |
| GCF_014844875.1_ASM1484487v1_genomic.fna | 0 | 0 | 0 | 1 | 1 | 0 | 1 | 1 | 1 | 1 | 1 | 0 |
| GCF_014844895.1_ASM1484489v1_genomic.fna | 0 | 0 | 0 | 1 | 1 | 0 | 1 | 1 | 1 | 1 | 1 | 0 |
| GCF_014844915.1_ASM1484491v1_genomic.fna | 0 | 0 | 0 | 1 | 1 | 0 | 1 | 1 | 1 | 1 | 1 | 0 |
| GCF_014844935.1_ASM1484493v1_genomic.fna | 0 | 0 | 0 | 1 | 1 | 0 | 1 | 1 | 1 | 1 | 1 | 0 |
| GCF_014844955.1_ASM1484495v1_genomic.fna | 0 | 0 | 0 | 1 | 1 | 0 | 1 | 1 | 1 | 1 | 1 | 0 |
| GCF_014844975.1_ASM1484497v1_genomic.fna | 0 | 0 | 0 | 1 | 1 | 0 | 1 | 1 | 1 | 1 | 1 | 0 |
| SRR7534671.fasta | 1 | 0 | 0 | 1 | 1 | 0 | 1 | 1 | 1 | 1 | 1 | 1 |
| SRR7534672.fasta | 0 | 0 | 0 | 1 | 1 | 0 | 1 | 1 | 1 | 1 | 1 | 0 |
| SRR7534673.fasta | 0 | 0 | 0 | 1 | 1 | 0 | 1 | 1 | 1 | 1 | 1 | 1 |
| SRR7534674.fasta | 0 | 0 | 0 | 1 | 1 | 0 | 1 | 1 | 1 | 1 | 1 | 0 |
| SRR7534675.fasta | 0 | 0 | 0 | 1 | 1 | 0 | 1 | 1 | 1 | 1 | 1 | 0 |
| SRR7534676.fasta | 1 | 0 | 0 | 1 | 1 | 0 | 1 | 1 | 1 | 1 | 1 | 0 |
| SRR7534677.fasta | 0 | 0 | 1 | 1 | 1 | 0 | 1 | 1 | 1 | 1 | 1 | 1 |
| SRR7534679.fasta | 0 | 0 | 0 | 1 | 1 | 0 | 1 | 1 | 1 | 1 | 1 | 1 |
| SRR7534680.fasta | 1 | 0 | 0 | 1 | 1 | 0 | 1 | 1 | 1 | 1 | 1 | 1 |
| SRR7534681.fasta | 0 | 0 | 0 | 1 | 1 | 0 | 1 | 1 | 1 | 1 | 1 | 1 |
| SRR7534682.fasta | 0 | 0 | 0 | 1 | 1 | 0 | 1 | 1 | 1 | 1 | 1 | 1 |
| SRR7534683.fasta | 1 | 0 | 0 | 1 | 1 | 0 | 1 | 1 | 1 | 1 | 1 | 0 |
| SRR7534685.fasta | 1 | 0 | 0 | 0 | 1 | 0 | 1 | 1 | 1 | 1 | 1 | 1 |
| SRR7534688.fasta | 1 | 0 | 0 | 1 | 1 | 0 | 1 | 1 | 1 | 1 | 1 | 1 |
| SRR7534689.fasta | 0 | 0 | 0 | 1 | 1 | 0 | 1 | 1 | 1 | 1 | 1 | 1 |
| SRR7534690.fasta | 1 | 0 | 0 | 1 | 1 | 0 | 1 | 1 | 1 | 1 | 1 | 1 |
| SRR7534691.fasta | 1 | 0 | 0 | 1 | 1 | 0 | 1 | 1 | 1 | 1 | 1 | 1 |
| SRR7534692.fasta | 1 | 0 | 0 | 1 | 1 | 0 | 1 | 1 | 1 | 1 | 1 | 1 |
| SRR7534693.fasta | 1 | 0 | 0 | 1 | 1 | 0 | 1 | 1 | 1 | 1 | 1 | 1 |
| SRR7534694.fasta | 1 | 0 | 0 | 1 | 1 | 0 | 1 | 1 | 1 | 1 | 1 | 1 |
| SRR7534695.fasta | 0 | 0 | 0 | 1 | 1 | 0 | 1 | 1 | 1 | 1 | 1 | 1 |
| SRR7534696.fasta | 0 | 0 | 0 | 1 | 1 | 0 | 1 | 1 | 1 | 1 | 1 | 1 |
| SRR7534697.fasta | 1 | 0 | 0 | 1 | 1 | 0 | 1 | 1 | 1 | 1 | 1 | 1 |
| SRR7534698.fasta | 0 | 0 | 0 | 1 | 1 | 0 | 1 | 1 | 1 | 1 | 1 | 1 |
| SRR7534699.fasta | 0 | 1 | 0 | 1 | 1 | 0 | 1 | 1 | 1 | 1 | 1 | 1 |

*(Continued)*

**Table 1.** (Continued)

| Sample | Number of respective genes in each sample | | | | | | | | | | | |
|---|---|---|---|---|---|---|---|---|---|---|---|---|
| | **TEM-1** | **TEM-104** | **TEM-206** | **farA** | **farB** | **lnuA** | **macA** | **macB** | **mtrC** | **mtrD** | **mtrE** | **tetM** |
| **SRR7534700.fasta** | 1 | 0 | 0 | 1 | 1 | 0 | 1 | 1 | 1 | 1 | 1 | 1 |
| **SRR7534701.fasta** | 0 | 0 | 0 | 1 | 1 | 0 | 1 | 1 | 1 | 1 | 1 | 0 |
| **SRR7534702.fasta** | 0 | 0 | 0 | 1 | 1 | 0 | 1 | 1 | 1 | 1 | 1 | 1 |
| **SRR7534703.fasta** | 0 | 0 | 0 | 1 | 1 | 0 | 1 | 1 | 1 | 1 | 1 | 0 |
| **SRR7534704.fasta** | 1 | 0 | 0 | 1 | 1 | 0 | 1 | 1 | 1 | 1 | 1 | 1 |
| **SRR7534705.fasta** | 0 | 1 | 0 | 1 | 1 | 0 | 1 | 1 | 1 | 1 | 1 | 1 |
| **SRR7534706.fasta** | 0 | 0 | 0 | 1 | 1 | 0 | 1 | 1 | 1 | 1 | 1 | 1 |
| **SRR7534707.fasta** | 1 | 0 | 0 | 1 | 1 | 1 | 1 | 1 | 1 | 1 | 1 | 1 |
| **SRR7534708.fasta** | 1 | 0 | 0 | 1 | 1 | 0 | 1 | 1 | 1 | 1 | 1 | 1 |
| **SRR7534709.fasta** | 0 | 0 | 0 | 1 | 1 | 0 | 1 | 1 | 1 | 1 | 0 | 1 |
| **SRR7534710.fasta** | 1 | 0 | 0 | 1 | 1 | 0 | 1 | 1 | 1 | 1 | 1 | 1 |

## Population genetic structure analysis

The population genetic structure was investigated through Principal Component Analysis (PCA) and hierarchical clustering, utilizing SNP genotype data derived from VCF files. The genotype data were numerically encoded to represent distinct alleles, and missing genotype values were imputed using column means to reduce potential bias. PCA was performed to reduce the dimensionality of the genotype data, enabling the visualization of genetic variation among samples in a two-dimensional space defined by the first two principal components (PC1 vs. PC2). The FST was calculated using vcftools to confirm the genetic differentiation between the clusters, which was visualized in python. The genetic relationships among the various genomes isolates was determined using a SNP distance matrix generated in PLINK version 1.9, then a hierarchical clustering based on Unweighted Pair Group Method with Arithmetic Mean (UPGMA). A minimum-spanning neighbor network of MDR-NG isolates was constructed from SNP pairwise distances. Edges represent the shortest set of connections linking all isolates without cycles, preserving overall distance relationships. Nodes are colored by sampling location.

## Genetic diversity analysis

Genetic diversity was evaluated using VCFtools and PLINK, with subsequent visualizations conducted in Python. Nucleotide diversity (π) was computed on a per-site basis using VCFtools, producing an output file containing π values (Table 2). To analyze genomic variation, the mean nucleotide diversity was calculated within 1000 bp sliding windows across the genome. The mean π value was also calculated and compared across geographic regions to assess regional differences in genomic heterogeneity. Linkage disequilibrium (LD) was assessed using PLINK. A BED file was generated from the genotype data, which served as the input for calculating pairwise LD (r² values) between SNPs. The resulting LD values were visualized using Python to investigate patterns of LD decay relative to genomic distances.

## The epidemiology of antimicrobial resistance (AMR) gene distribution and statistical analysis

The distribution of AMR gene counts per isolate across the five geographic regions (Coastal Region, Kisumu, Kombewa, Nairobi, and Rift Valley) were compaired. With the multiple comparison groups and the observed non-normal distribution of AMR gene counts, a non-parametric Kruskal-Wallis H test was employed to test the median AMR gene distribution was equal across all regions. The specific regional pairs were identified, using the Dunn's post-hoc- test. All pairwise comparisons utilized the Bonferroni correction to adjust for multiple testing and

**Table 2. Nucleotide diversity average values per sample. The average nucleotide diversity values distributed per sample genome.**

| Sample | Average Nucleotide Diversity (π) |
|---|---|
| GCA_014844795.1_ASM1484479v1_genomic.fna | 0.219739 |
| GCA_014844815.1_ASM1484481v1_genomic.fna | 0.201697 |
| GCA_014844835.1_ASM1484483v1_genomic.fna | 0.24018 |
| GCA_014844855.1_ASM1484485v1_genomic.fna | 0.226343 |
| GCA_014844875.1_ASM1484487v1_genomic.fna | 0.181847 |
| GCA_014844895.1_ASM1484489v1_genomic.fna | 0.197271 |
| GCA_014844915.1_ASM1484491v1_genomic.fna | 0.19097 |
| GCA_014844935.1_ASM1484493v1_genomic.fna | 0.213601 |
| GCA_014844955.1_ASM1484495v1_genomic.fna | 0.187259 |
| GCA_014844975.1_ASM1484497v1_genomic.fna | 0.216502 |
| GCF_014844795.1_ASM1484479v1_genomic.fna | 0.219739 |
| GCF_014844815.1_ASM1484481v1_genomic.fna | 0.201697 |
| GCF_014844835.1_ASM1484483v1_genomic.fna | 0.24018 |
| GCF_014844855.1_ASM1484485v1_genomic.fna | 0.226343 |
| GCF_014844875.1_ASM1484487v1_genomic.fna | 0.181847 |
| GCF_014844895.1_ASM1484489v1_genomic.fna | 0.197271 |
| GCF_014844915.1_ASM1484491v1_genomic.fna | 0.19097 |
| GCF_014844935.1_ASM1484493v1_genomic.fna | 0.213601 |
| GCF_014844955.1_ASM1484495v1_genomic.fna | 0.187259 |
| GCF_014844975.1_ASM1484497v1_genomic.fna | 0.216502 |
| SRR7534671 | 0.133386 |
| SRR7534672 | 0.143251 |
| SRR7534673 | 0.142902 |
| SRR7534674 | 0.089783 |
| SRR7534675 | 0.154975 |
| SRR7534676 | 0.157368 |
| SRR7534677 | 0.14378 |
| SRR7534679 | 0.143306 |
| SRR7534680 | 0.1435 |
| SRR7534681 | 0.150355 |
| SRR7534682 | 0.120512 |
| SRR7534683 | 0.144555 |
| SRR7534685 | 0.144875 |
| SRR7534688 | 0.110644 |
| SRR7534689 | 0.145777 |
| SRR7534690 | 0.148583 |
| SRR7534691 | 0.096767 |
| SRR7534692 | 0.096189 |
| SRR7534693 | 0.154522 |
| SRR7534694 | 0.104211 |
| SRR7534695 | 0.146742 |
| SRR7534696 | 0.141247 |
| SRR7534697 | 0.10273 |
| SRR7534698 | 0.130315 |
| SRR7534699 | 0.0967 |

*(Continued)*

**Table 2.** (Continued)

| Sample | Average Nucleotide Diversity (π) |
|---|---|
| SRR7534700 | 0.107115 |
| SRR7534701 | 0.11462 |
| SRR7534702 | 0.156684 |
| SRR7534703 | 0.19132 |
| SRR7534704 | 0.149096 |
| SRR7534705 | 0.113442 |
| SRR7534706 | 0.155605 |
| SRR7534707 | 0.098244 |
| SRR7534708 | 0.103165 |
| SRR7534709 | 0.152193 |
| SRR7534710 | 0.107218 |

control the False Discovery Rate. The prevalence of individual AMR gene and specific antibiotic resistance across the study geographic regions were determined. A heatmap was generated to visually depict the regional distribution of the prevalence of key AMR genes. Furthermore, the identified AMR genes were summarized by their corresponding antibiotic drug class (e.g., β-lactam, macrolide, tetracycline, efflux pump/MDR) to characterize the regional clustering of resistance mechanisms.

## Results

### Processing and determination of variants

Variant calling process across all 56 genomes yielded a total of 50710 Single Nucleotide Polymorphisms (SNPS) (Fig 1).

### Population genetic structure analysis

Our analysis of the population genetic structure revealed distinct clustering patterns (Fig 2). The principal components (PC1 and PC2) shows proportion of the total genetic variation, with PC1 67.59% and PC2 4.44% of the variance (Fig 2). The genomes outside the primary cluster, includes GCA_014844855.1, GCA_014844815.1, GCA_014844795.1, GCA_014844975.1, GCA_014844895.1, GCA_014844935.1, GCA_014844955.1, GCA_014844875.1, and GCA_014844915.1 which are all from Nairobi. The genome of others isolates shows close genetic relationships within the primary cluster. The FST statistics indicated that most genomic regions had low FST ranging from 0 to 0.1 while it also showed distinct peaks of high FST of up to 0.8–0.9 (Fig 3). The Neighbor-Net network analysis revealed the genetic structure among the genome of the isolates (Fig 4). The split network showed a dominant, well-connected cluster, aligning with the clustering patterns in the PCA plot.

### Nucleotide diversity (π) and patterns of LD

The nucleotide diversity (π) plot showed variations in genetic diversity across all genomes, with π values ranging from 0.0 to ~ 0.6 (Fig 5). Multiple peaks were observed at distinct genomic positions, indicating regions of notable genetic variation. Regional comparisons showed that isolates from Kisumu and Kombewa exhibited significantly higher mean π values (diversity) than those from Nairobi and the Coast (Fig 6). The LD decay plot reveals the relationship between LD (measured as $r^2$) and genomic distance and the LD values and genomic distance were inversely proportional (Fig 7). The fitted decay curve (represented by the red line) had a rapid decline in LD within the first 1000 base pairs (bp), then a gradual decrease which stabilized beyond 2000 bp (Fig 7).

 

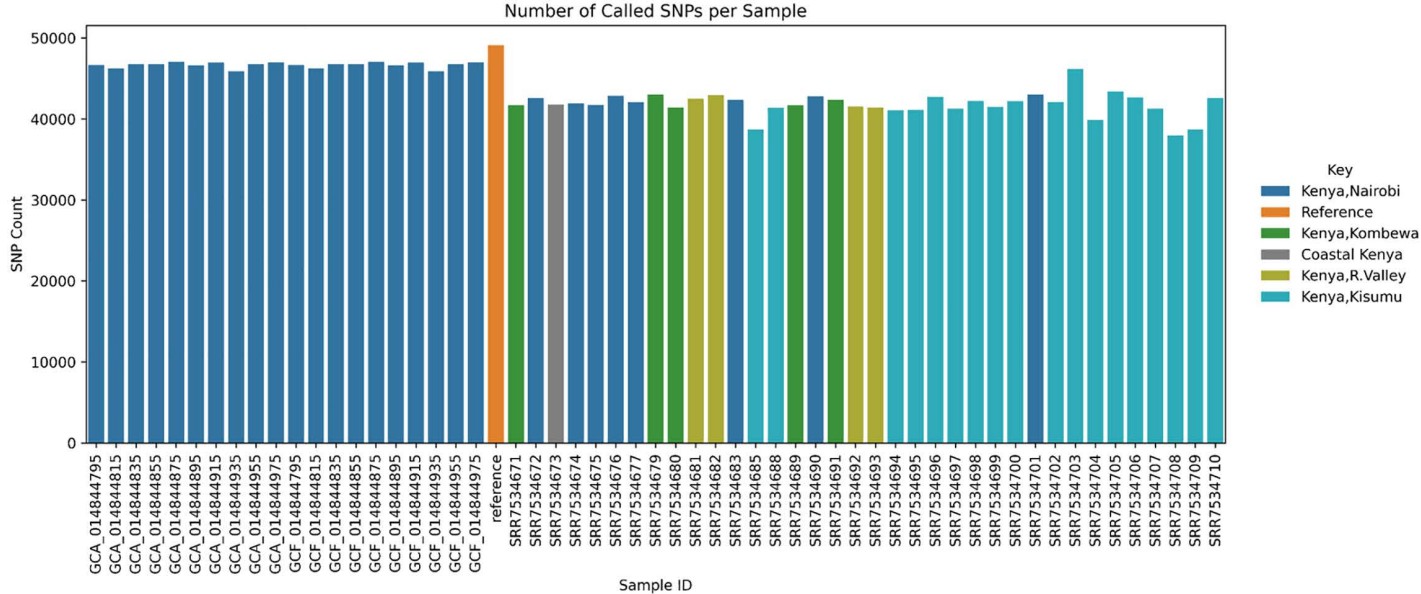

**Fig 1. SNP count per sample.** Number of SNPs identified across each of the 56 *N. gonorrhoeae* isolates with color code.

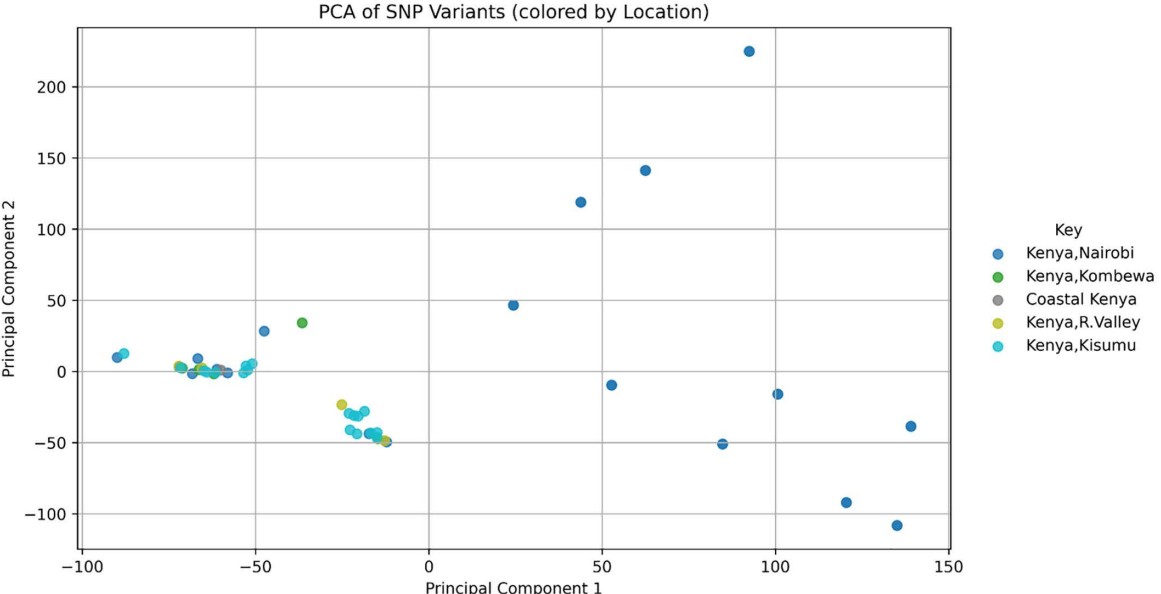

**Fig 2. Principal component analysis (PCA) of MDR-NG isolates from Kenya.** Clustering pattern showing one dominant group and several divergent outliers. The samples are coded with their location.

## The epidemiology of antimicrobial resistance (AMR) gene distribution and statistical analysis

The Antimicrobial Resistance (AMR) gene burden per isolate showed significant variation across the geographic regions (Fig 8). The non-parametric Kruskal–Wallis H test indicated an overall significant difference in median AMR gene distribution (H = 31.36, p < 0.0001).

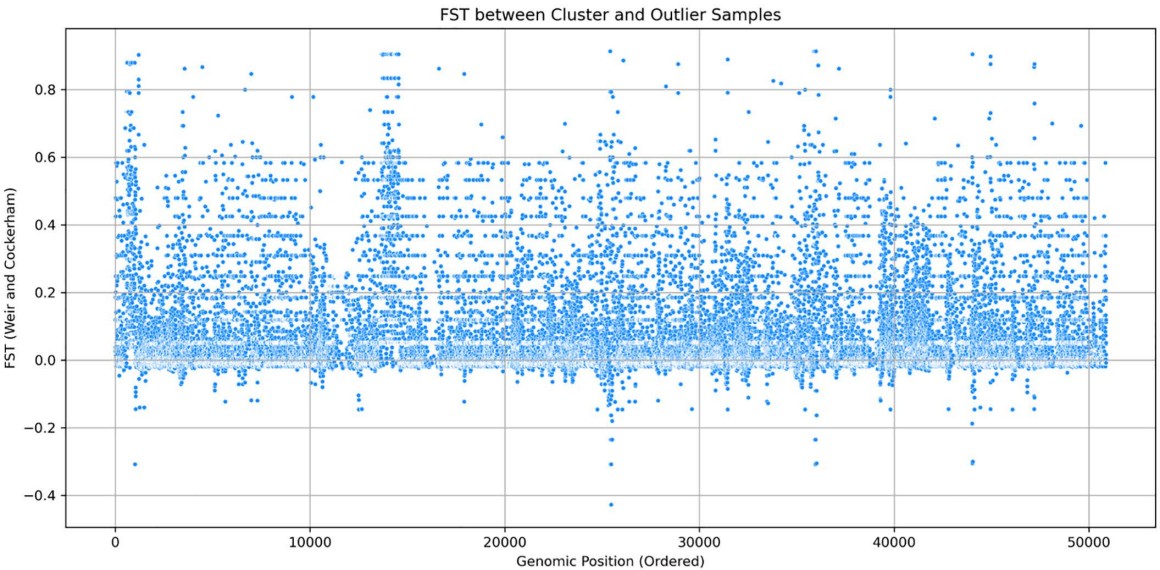

**Fig 3. FST analysis between PCA cluster and outliers.** Scatter plot of FST between cluster and outliers of PCA.

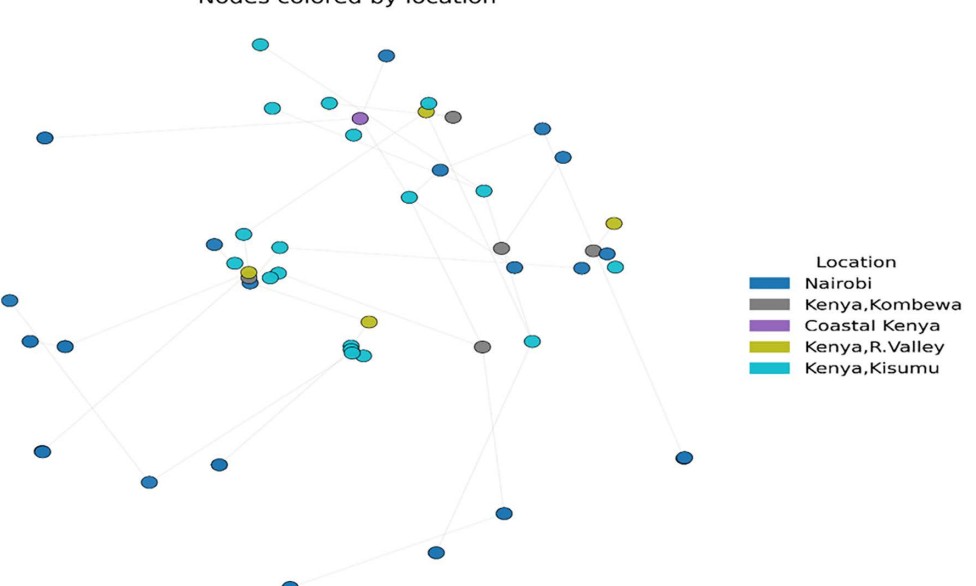

**Fig 4. Neighbor network analysis of MDR-NG isolates from Kenya.** A minimum-spanning neighbor network of MDR-NG isolates, constructed from SNP pairwise distances. Edges represent the shortest set of connections linking all isolates without cycles, preserving overall distance relationships. Nodes are colored by sampling location.

Post-hoc analysis using the Dunn's test revealed specific regional clustering of high resistance burden. Isolates from Kisumu, Kombewa, and the Rift Valley harbored significantly more AMR genes compared to those from Nairobi (Fig 9). For example, the mean AMR gene counts per isolate were highest in Kombewa and Rift Valley, and lowest in Nairobi and the Coastal Region.

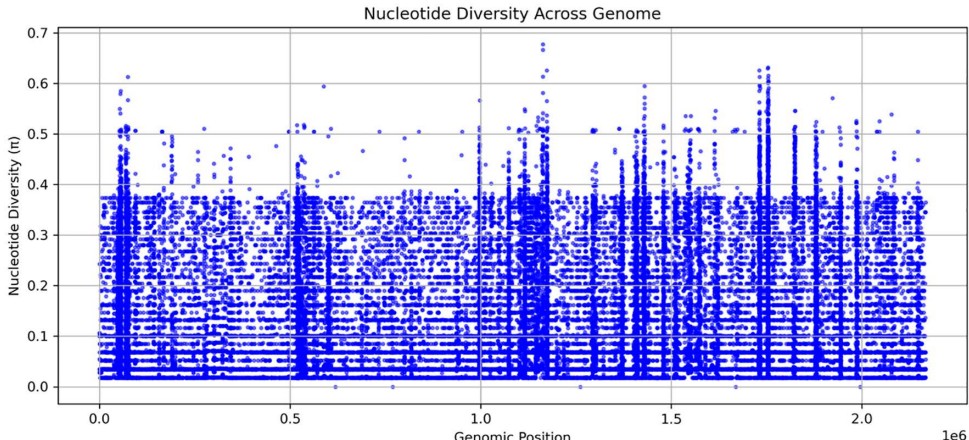

**Fig 5. Nucleotide diversity (π) across the genome.** Nucleotide diversity across the genomes with peaks indicating regions of high genetic variability.

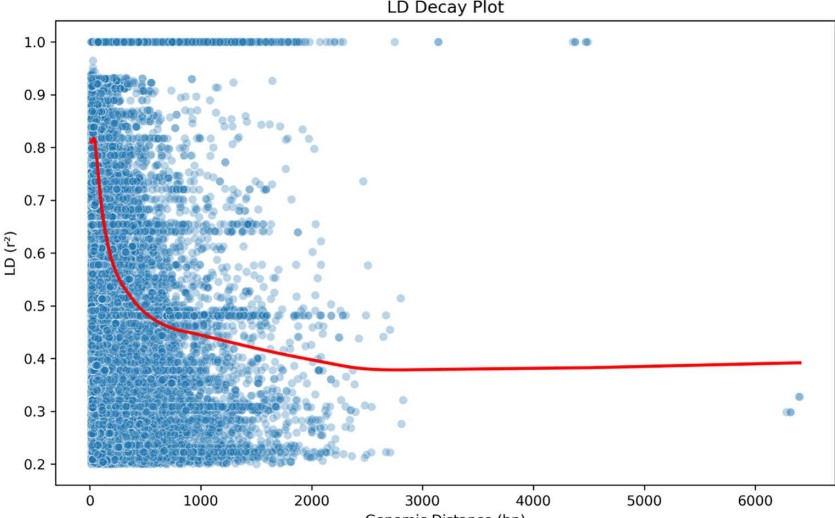

**Fig 6. Nucleotide diversity (π) per region.** Box plot showing the mean nucleotide diversity (π) per isolate for each geographic region, illustrating regional differences in genomic heterogeneity.

The analysis of individual resistance genes revealed two distinct patterns: ubiquitous core mechanisms and regionally restricted specific determinants. Core resistance genes—including those encoding efflux pumps (farA, farB, mtrC, mtrD, mtrE) and macrolide resistance (macA, macB)—were nearly universally present (>93% prevalence) across all geographic regions (Fig 9). The tetracycline resistance gene tetM was also widespread but exhibited markedly lower prevalence in the Coastal region. In contrast, β-lactamase genes demonstrated significant regional clustering. TEM-type variants (TEM-1, TEM-104, TEM-206) were predominantly found in Western Kenya. The TEM-1 gene was highly prevalent in Kisumu (60%) and Kombewa (30%) but absent in Nairobi and the Coast (Fig 10). Other genes, including lnuA and TEM-206, were detected at low frequencies and in specific regions only.

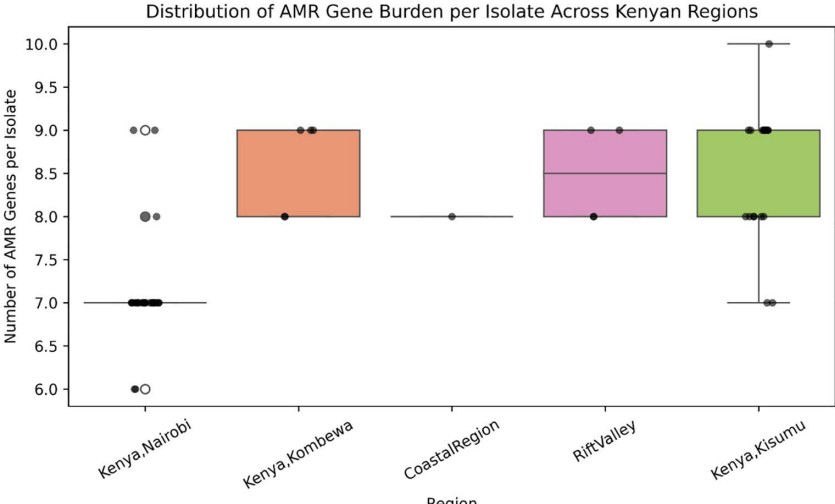

**Fig 7. Linkage disequilibrium (LD) decay plot for MDR-NG isolates.** Rapid LD decay within 1,000 bp reflects high recombination rates. Plateau beyond 2,000 bp indicates stabilized allele associations.

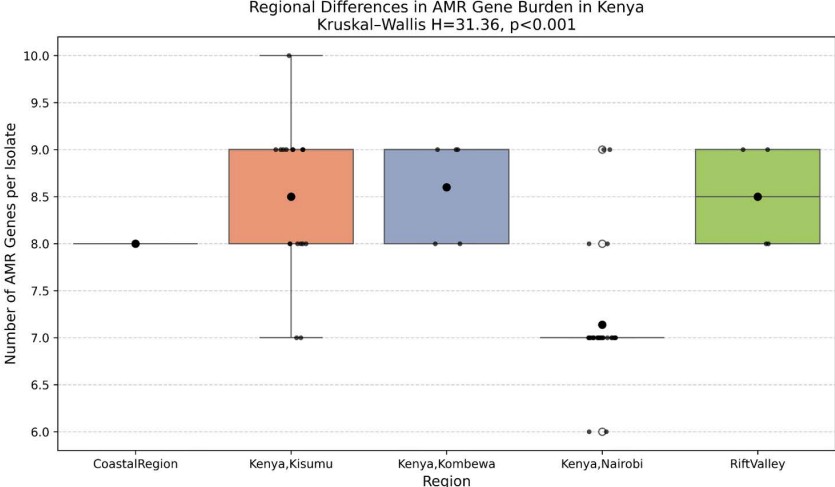

**Fig 8. Distribution of AMR gene burden per isolate across kenyan regions.** Box plot showing the distribution of the total number of Antimicrobial Resistance (AMR) genes per isolate across five geographic regions.

## Discussion

This study offers a detailed analysis of the genetic population structure and diversity of multidrug-resistant *Neisseria gonorrhoeae* isolates from Kenya using genome-wide SNP analysis. Our findings provide valuable insights informing targeted treatment strategies to combat *Neisseria gonorrhoeae* infections within the Kenyan population. Key observations included genetic differentiation, evidence of recombination, and critical insights into the mechanisms underlying antibiotic resistance evolution. These findings highlight the transmissibility and persistence of *Neisseria gonorrhoeae* in human populations, hence need for tailored interventions to address the growing challenge of antimicrobial resistance.

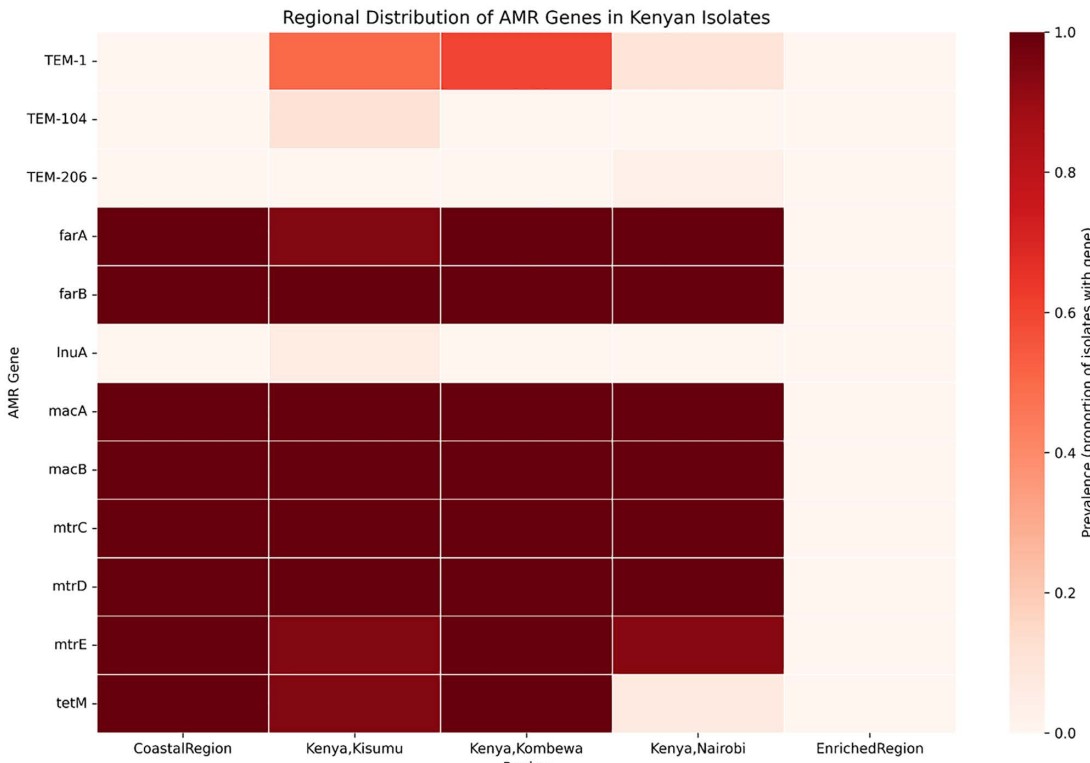

**Fig 9. Regional differences in AMR gene burden in Kenya.** Box plot illustrating the number of AMR genes per isolate across the five geographic regions, showing the result of the Kruskal-Wallis H test.

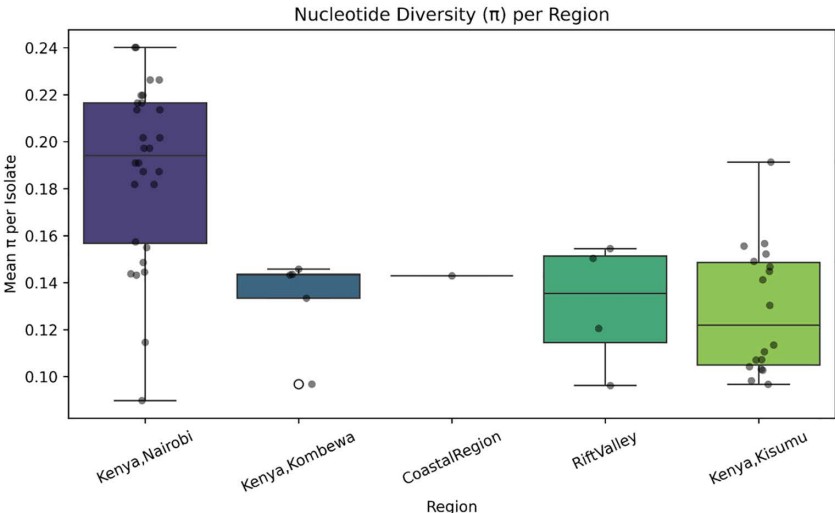

**Fig 10. Regional distribution of specific AMR genes in Kenyan isolates.** Heatmap depicting the prevalence (proportion of isolates with gene) of key AMR genes across the five geographic regions.

The PCA revealed distinct clustering patterns, showing the presence of genetically differentiated subpopulations which the Kenyan *N gonorrhoeae* isolates. A substantial portion of the genetic variation was summarized along the first two principal components, suggesting distinct patterns in the data. A predominant cluster encompassing most isolates, alongside a few genetically divergent outliers. This points to limited overall diversity with potential signals of distinct evolutionary origins or introductions from external sources. Despite these clusters, no strong geographic structure was observed, suggesting that genetic variation is not entirely location-dependent but could be shaped by other selective pressures, such as antibiotic use and recombination events. The absence of clear geographic structuring is consistent with the high mobility of *Neisseria gonorrhoeae* and its ability to spread across populations due to human migration and sexual behaviors [16].

To investigate further the genetic differentiation between the main genetic cluster in the PCA results and the outlier samples indicated in the PCA plot (Fig 2), a genome-wide FST analysis was conducted using Vcftools. The results (Fig 3) revealed that while the majority of the loci showed a low genetic differentiation, represented by the regions with FST 0–0.1, several genomic regions displayed an elevated FST of 0.4. The peaks of the FST analysis plot reach as high as 0.8, which indicates that there are regions which are under divergent selection or horizontal gene transfer events, and may be driving the observed structure in the PCA (Fig 2).

The Neighbor-Net analysis further supported our findings by revealing a well-connected primary cluster with several reticulations in the network. These reticulations suggest recombination events and shared genetic ancestry between some isolates. The presence of longer branch lengths in certain isolates supports the idea of genetic divergence that may be associated with acquired resistance mutations [17].

The findings on AMR gene distributions and prevalence provide an essential molecular epidemiological correlation to the spread of *N. gonorrhoeae* multidrug resistance in Kenya. The observed significant difference in the AMR gene, with isolates from Western Kenya (Kisumu, Kombewa) had significantly more resistance genes than those from the Central region (Nairobi) (Fig 7), suggests the existence of regional hotspots of MDR. This is a reflection of differint antibiotic usage patterns, treatment practices, and sexual behaviour to these geographic areas.The high prevalence of core efflux pump and macrolide resistance genes across all regions confirms that widespread, baseline resistance mechanisms beingendemic in Kenya. However, the observed regional concentration of TEM-type β-lactamase variants in Western Kenya is of great concern. This indicates that while resistance mechanisms like the efflux pumps offer general drug resistance, specific resistance genes, such as those conferring resistance to β-lactams, may be geographically localized due to recent selective pressures or clonal expansion. This authenticate molecular evidence in supporting clinical surveillance reports that often suggest higher rates of treatment failure or elevated minimum inhibitory concentrations (MICs) in the western part of the country.

Extensive evidence of recombination in the Kenyan N gonorrhoeae isolates is a key finding of our study. Linkage disequilibrium decay analysis revealed frequent recombination events breaking down genetic associations over increasing distance [18]. High LD values (~0.8–1.0) at ~0 bp suggest tight linkage from low recombination, but LD decays rapidly by ~500–2000 bp due to increased recombination. Beyond 2000–3000 bp, LD stabilized at lower levels (~0.3–0.4), suggesting that longer genomic distances are more frequently broken by recombination events, resulting in a random association of alleles. The observed rapid LD decay pattern provides strong evidence that recombination is a dominant force shaping genetic variation in MDR *Neisseria gonorrhoeae* Our results are consistent with findings by Manoharan-Basil et al.[19], who reported that *Neisseria gonorrhoeae* undergoes frequent horizontal gene transfer, facilitating the acquisition of beneficial mutations, particularly those associated with antibiotic resistance [20].

Analysis of the nucleotide diversity (π) supports the role of recombination in shaping genetic variation, as regions with high nucleotide diversity overlapped with those exhibiting low LD. These findings suggest that recombination actively contributes to the genetic diversification of *Neisseria gonorrhoeae* populations in Kenya. The demonstration of highest genomic diversity in Kisumu and Kombewa aligns directly with the highest observed AMR gene in these regions. High π values suggest a population subject to greater recombination, frequent introduction of new strains, or multiple

 

co-circulating lineages [21]. Recombination hotspots in the genome are reflected as peaks of high nucleotide diversity in the scatter plot, suggesting a potential role in driving the adaptive evolution of antimicrobial resistance. This pattern is critical for the bacterium, as it enables rapid evolution in response to selective pressures, such as antibiotic use. In contrast, the lower diversity observed in Nairobi and the Coastal Region suggests more clonal populations. While these regions still carry core resistance genes, the lower diversity may reflect a scenario where a few successful, dominant clones are responsible for most transmission, or simply that the sampling captured a more limited genetic breadth. These regional differences in genomic heterogeneity underscore the importance of localized surveillance efforts to tailor public health interventions against *N. gonorrhoeae* effectively. The findings underscore that recombination, in combination with high mutation rates, plays a central role in facilitating the adaptive potential of *Neisseria gonorrhoeae.* Unlike many other bacteria that primarily rely on clonal expansion, *Neisseria gonorrhoeae* possesses a highly dynamic genome, allowing it to survive and spread even with ongoing treatment. There is a strong interplay between recombination and genetic variability, which results in the evolutionary adaptation of the organism [22].

The high recombination rates and genetic diversity observed in *Neisseria gonorrhoeae* have significant implications for its transmissibility and evolution of antibiotic resistance [23]. The genetic exchange and dissemination are pertinent in influencing genetic diversity and high recombination rates. The clusters that are observed in the population structure analysis indicate *Neisseria gonorrhoeae* can rapidly transfer resistance mutations within the population and across populations. The genetic flexibility of *Neisseria gonorrhoeae* promotes the spread of resistance to antibiotics such as tetracycline, penicillin, and cephalosporins, largely due to its high adaptability [24]. Recombination events contribute to increased transmissibility and complicate infection control by enhancing the pathogen's fitness and survival across diverse populations. This adaptability facilitates easier transmission between hosts and supports the persistence of multidrug-resistant *Neisseria gonorrhoeae* strains, making eradication efforts increasingly challenging [5]. *Neisseria gonorrhoeae* readily adapts to new hosts, and its genetic composition and diversity indicate that it is easily transmitted across diverse populations [25]. These findings underscore the necessity for more targeted antibiotic strategies that consider the genetic diversity and population structure of the bacterium, rather than approaching it as a homogeneous entity. Insights into the population structure reveal how adaptive mechanisms have contributed to the emergence of new *Neisseria gonorrhoeae* strains with diverse genetic compositions associated with antimicrobial resistance. High genomic plasticity of *Neisseria gonorrhoeae* complicates current treatment regimens as new resistant strains can emerge unpredictably, which highlights the urgent need for adaptive treatment strategies, including combination therapies and genomic surveillance-based interventions, to effectively manage and control the spread of resistant gonococcal strains.

## Conclusion

This study advances the understanding of antibiotic resistance in *Neisseria gonorrhoeae* by characterizing its genetic diversity and population structure through SNP analysis, and by providing new insights into its recombination dynamics. The LD analysis revealed a rapid decay over short genomic distances, indicating high recombination activity among the isolates. This recombination contributes to elevated genetic diversity and facilitates the spread of antimicrobial resistance traits. Consistently, nucleotide diversity peaks aligned with regions of rapid LD decay, reinforcing the conclusion that recombination is a key driver of resistance evolution. The findings of the nucleotide diversity has shown that high recombination rates observed in the genomes of *Neisseria gonorrhoeae* are the key drivers of antimicrobial resistance, which increases its transmissibility. The analysis of the population structure of *Neisseria gonorrhoeae* from Kenyan isolates revealed different populations, although with outliers, indicating an ongoing development of resistance strains.

We confirm that Kenyan *N. gonorrhoeae* isolates are predominantly multidrug-resistant (MDR), characterized by the widespread presence of core resistance mechanisms, notably efflux pump genes (mtrCDE, farAB) and macrolide resistance determinants (macAB).The molecular epidemiology analysis identified significant regional heterogeneity that impacts public health strategy. The AMR gene distribution is not uniform, with isolates from Western Kenya (Kisumu,

Kombewa, Rift Valley) harboring a significantly higher number of resistance genes and reflecting genetic diversity (higher π values) compared to the more clonal populations in the Central and Coastal regions. This suggests that Western Kenya may act as a hotspot for genetic exchange, facilitating the spread of antimicrobial resistance traits,thusestablishing high genetic diversity and AMR gene positively correlating in these regions.

The ongoing global spread of *N. gonorrhoeae* and the potential for co-infections with other emerging pathogens, our findings are pertinent in informing public health policy. The non-uniform distribution of resistance mandates a shift towards localized surveillance and regionally tailored treatment guidelines. Future control efforts should prioritize enhanced molecular surveillance in notable diversity, high resistance burden regions like Western Kenya to monitor the emergence of novel resistance types and ensure the continued efficacy of first-line therapies.

## Supporting information

**S1 File. Busco completeness score.**
(DOCX)

## Author contributions

**Conceptualization:** Dennis Wamalabe Mukhebi.

**Data curation:** Henry Kiema Musee.

**Formal analysis:** Henry Kiema Musee.

**Investigation:** Dennis Wamalabe Mukhebi.

**Methodology:** Henry Kiema Musee.

**Project administration:** Dennis Wamalabe Mukhebi.

**Supervision:** Dennis Wamalabe Mukhebi.

**Validation:** Henry Kiema Musee.

**Visualization:** Henry Kiema Musee.

**Writing – review & editing:** Dennis Wamalabe Mukhebi.

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
