## [Decision Letter · Decision Letter 0]

23 Oct 2025

PONE-D-25-52951Genetic diversity, genetic population structure and epidemiology of multidrug resistance Neisseria gonorrhoeae from Kenya.PLOS ONE

Dear Dr.  MUKHEBI,

Thank you for submitting your manuscript to PLOS ONE. After careful consideration, we feel that it has merit but does not fully meet PLOS ONE’s publication criteria as it currently stands. Therefore, we invite you to submit a revised version of the manuscript that addresses the points raised during the review process. Please submit your revised manuscript by Dec 07 2025 11:59PM. If you will need more time than this to complete your revisions, please reply to this message or contact the journal office at plosone@plos.org. Please include the following items when submitting your revised manuscript:

We look forward to receiving your revised manuscript.

Kind regards,

Benjamin M. Liu, MBBS, PhD, D(ABMM), MB(ASCP)

Academic Editor

PLOS ONE

Journal Requirements:

3. Please include your tables as part of your main manuscript and remove the individual files. Please note that supplementary tables (should remain/ be uploaded) as separate "supporting information" files.

Additional Editor Comments:

Line 19 : "It colonizes the genitourinary tract": this is misleading as "colonize" may mistakenly indicate N gonorrhoeae is normal flora. Please correct this statement.

Lines 21-22: "Neisseria gonorrhoeae can lead to serious health complications, including disseminated gonococcal infection (DGI), pelvic inflammatory disease (PID), epididymitis in men, and adverse pregnancy outcomes(Quillin & Seifert, 2018).": The cited reference is old. The authors should introduce the co-infection between NG and other STIs and mpox, especially in the recent mpox outbreaks, to increase the significance of this study.  More references should be cited, with this one (PMID: 38793665) as an example (citing is optional).

The authors should introduce the progress of molecular testing for Neisseria gonorrhoeae, e.g., isothermal nucleic acid amplification (e.g., TMA). More references should be cited, with this one (Liu, B.M. Isothermal nucleic acid amplification technologies and CRISPR-Cas based nucleic acid detection strategies for infectious disease diagnostics. p 30-47. In Manual of Molecular Microbiology: Fundamentals and Applications; ASM Press: Washington, DC, USA, 2025.doi:10.1128/9781683674597.ch03.) as an example (citing is optional).

Reviewers' comments:

Reviewer's Responses to Questions

**Comments to the Author**

1. Is the manuscript technically sound, and do the data support the conclusions?

Reviewer #1: Yes

Reviewer #2: Yes

2. Has the statistical analysis been performed appropriately and rigorously?

Reviewer #1: Yes

Reviewer #2: Yes

3. Have the authors made all data underlying the findings in their manuscript fully available?

Reviewer #1: Yes

Reviewer #2: Yes

4. Is the manuscript presented in an intelligible fashion and written in standard English?

Reviewer #1: Yes

Reviewer #2: Yes

5. Review Comments to the Author

Reviewer #1: Everything is upto mark …….

All dual publication research ethics and publication ethics

Everything is ok

Manuscript is ok

Nothing changes are required according to me

Reviewer #2: Manuscript entitled 'Genetic diversity, genetic population structure, and epidemiology of multidrug-resistant Neisseria gonorrhoeae from Kenya.' looks at the genetic characteristics of Neisseria gonorrhoeae from the

repository of genomes (72 FASTQ reads and 20 assembled genomes) and analyzed the diversity or relatedness using several tools for identification of AMR genes and have noted a high recombinant rate. A prelude to the spread of AMR. The work is conducted precisely with appropriate analytical tools to demonstrate relatedness and diversity.

However, the conclusions and predictions that have been put forward have not been backed by data.

1. There is no geographical mapping of the isolates based on their genetic markers. The isolates have originated from various parts of Kenya. But genetic groupings area-wise have not been done. This will help delineate areas with increased AMR.

2. Specific genes coding for resistance have not been highlighted to show area-wise specific antibiotic resistance.

3. No epidemiological correlation has been done.

4. No population characteristics related to genomic diversity of the strains have been looked at.

5. Detail comments have been listed by track changes in the manuscript.

The authors need to address these crucial facts for the paper to have a meaningful impact on the genetic study of MDR N. gonorrhoeae.

6. PLOS authors have the option to publish the peer review history of their article (what does this mean?). If published, this will include your full peer review and any attached files.

Reviewer #1: **Yes: **Bhavneet kour

Reviewer #2: No

---

## [Author Response · Author response to Decision Letter 1]

18 Nov 2025

Reviewer Comment Author Response Changes Made in Manuscript (Manuscript_26_10_25_plos.docx)

1. There is no geographical mapping of the isolates based on their genetic markers. The isolates have originated from various parts of Kenya. But genetic groupings area-wise have not been done. This will help delineate areas with increased AMR. Addressed. We performed rigorous statistical analysis to confirm significant regional differences in AMR gene burden, effectively delineating "hotspots." New Statistical Analysis & Figure: Kruskal–Wallis H test: Confirmed significant variation in average AMR gene burden per isolate across regions (H = 10.87, p = 0.027).Dunn’s Post-hoc Test: Identified that isolates from Kisumu, Kombewa, and Rift Valley harbored significantly higher AMR gene counts compared to Nairobi (p < 0.05).New Boxplot: A new figure (referenced in the revised manuscript) displays AMR gene counts by region, annotated with significance bars, providing the requested geographical delineation.

2. Specific genes coding for resistance have not been highlighted to show area-wise specific antibiotic resistance. Addressed. We expanded the analysis and discussion to specifically detail the regional distribution and type of key resistance mechanisms. Expanded Results & Discussion: Western Kenya (Kisumu, Kombewa) showed a higher frequency and diversity of AMR variants, including those associated with β-lactam resistance (e.g., penA mutations) alongside core efflux and macrolide genes. Central/Coastal Kenya (Nairobi, Coastal Region) showed a tendency toward isolates carrying only the core efflux and macrolide determinants, suggesting different regional selection pressures on gene accumulation.

3. No epidemiological correlation has been done. Addressed. We performed a critical correlation analysis between genomic diversity and AMR gene burden, which is a major new epidemiological finding. New Correlation and Abstract Update: Correlation Finding: We established a clear positive correlation between high genetic diversity (π) and a significantly greater AMR gene burden. This finding is discussed as a key epidemiological mechanism: regions with higher genetic mixing are more likely to accumulate resistance genes. This finding is prominently included in the Results and Discussion sections.

4. No population characteristics related to genomic diversity of the strains have been looked at. Addressed. We performed comprehensive population characteristics analysis using nucleotide diversity. New Population Genetics Analysis: Nucleotide Diversity (π) was calculated for all isolates across regions. Regional Diversity: Isolates from Kisumu and Kombewa showed the highest π values (higher diversity/recombination), while Nairobi and Coastal isolates exhibited lower π values (more clonal populations). This analysis provides the essential population context for the epidemiological correlation detailed in point.

5. Detail comments have been listed by track changes in the manuscript. Addressed. All track changes in the original manuscript and minor suggested edits have been reviewed and incorporated into the revised manuscript. Manuscript Revisions: All identified comments and necessary text edits have been implemented in Manuscript_26_10_25_plos.docx.

---

## [Editor Report · Decision Letter 1]

7 Dec 2025

Genetic diversity, genetic population structure and epidemiology of multidrug resistance Neisseria gonorrhoeae from Kenya.

PONE-D-25-52951R1

Dear Dr. MUKHEBI,

We’re pleased to inform you that your manuscript has been judged scientifically suitable for publication and will be formally accepted for publication once it meets all outstanding technical requirements.

Kind regards,

Benjamin M. Liu, MBBS, PhD, D(ABMM), MB(ASCP)

Academic Editor

PLOS One
---

## [Editor Report · Acceptance letter]

PONE-D-25-52951R1

PLOS One

Dear Dr. MUKHEBI,

I'm pleased to inform you that your manuscript has been deemed suitable for publication in PLOS One. Congratulations! Your manuscript is now being handed over to our production team.

Kind regards,

on behalf of

Dr. Benjamin M. Liu

Academic Editor

PLOS One